# A Review of the Methods of Non-Invasive Assessment of Intracranial Pressure through Ocular Measurement

**DOI:** 10.3390/bioengineering9070304

**Published:** 2022-07-11

**Authors:** Jinhui Dong, Qi Li, Xiaofei Wang, Yubo Fan

**Affiliations:** Key Laboratory for Biomechanics and Mechanobiology of Ministry of Education, Beijing Advanced Innovation Center for Biomedical Engineering, School of Biological Science and Medical Engineering, Beihang University, Beijing 100083, China; jhdong@buaa.edu.cn (J.D.); by2010123@buaa.edu.cn (Q.L.)

**Keywords:** cerebrospinal fluid pressure, ocular measurement, non-invasive, optic nerve, biomechanics

## Abstract

The monitoring of intracranial pressure (ICP) is essential for the detection and treatment of most craniocerebral diseases. Invasive methods are the most accurate approach to measure ICP; however, these methods are prone to complications and have a limited range of applications. Therefore, non-invasive ICP measurement is preferable in a range of scenarios. The current non-invasive ICP measurement methods comprise fluid dynamics, and ophthalmic, otic, electrophysiological, and other methods. This article reviews eight methods of non-invasive estimation of ICP from ocular measurements, namely optic nerve sheath diameter, flash visual evoked potentials, two-depth transorbital Doppler ultrasonography, central retinal venous pressure, optical coherence tomography, pupillometry, intraocular pressure measurement, and retinal arteriole and venule diameter ratio. We evaluated and presented the indications and main advantages and disadvantages of these methods. Although these methods cannot completely replace invasive measurement, for some specific situations and patients, non-invasive measurement of ICP still has great potential.

## 1. Introduction

Intracranial pressure (ICP) is the pressure within the cerebrospinal compartment, which comprises a fixed volume of neural tissue, cerebrospinal fluid (CSF), and blood. The normal range of ICP is 5–15 mmHg for healthy adults [1]. The Monro–Kellie doctrine states that the total volume of neural tissue, CSF, and blood is constant [2]. Increases in the volume of the contents of the cerebrospinal compartment, such as that which occurs with brain edema, increased blood volume, hydrocephalus, and intracranial space-occupying lesions, leads to an increase in ICP. Elevated ICP causes a series of clinical symptoms, such as headache, nausea, and vomiting. ICP measurement is vital for the monitoring and treatment of craniocerebral diseases. ICP measurement is used in neurology and neurosurgery and also has a wide range of applications in ophthalmology and aerospace medicine. In ophthalmology, both increases and decreases in ICP are associated with optic nerve disease. Elevated ICP can lead to papilledema, and decreased ICP may be a pathophysiological factor in the development of glaucoma [3,4].

Currently, invasive ICP measurement is often used in clinical practice. In 1960, Lundberg achieved invasive continuous monitoring of ICP [5]. Since then, invasive ICP measurement has developed dramatically. External ventricular drainage is a routine clinical monitoring method that allows drainage of CSF via a catheter placed in the ventricle, and CSF pressure is measured by an external pressure sensor. This technology is considered the gold standard for ICP measurement. The resulting ICP value is accurate, and ICP can be controlled by draining the CSF. However, this method is associated with numerous complications, such as intracranial infection. The use of fiber optic devices to monitor ICP is also widely used in clinical practice. This method of monitoring can be performed in the intraventricular, intraparenchymal, epidural, or subdural space [6]. In addition, the implantation of miniature wireless sensors in the brain parenchyma also enables continuous monitoring of a patient’s ICP [7]. Lumbar puncture can be used to measure ICP for patients with communicating CSF pathways, and is one of the most widely used methods in clinical practice. Although invasive ICP monitoring requires dedicated space/environment and is prone to associate with some complications, it still has an irreplaceable role for many patients. For example, for patients with severe craniocerebral injury, dynamic ICP monitoring is an important indicator to determine or adjust treatment plans. Because it is difficult to achieve ideal accuracy and technological requirements for continuous monitoring with non-invasive ICP measurement equipment, invasive ICP measurement is still the first choice for monitoring ICP in patients with severe traumatic brain injury (TBI).

Although invasive methods are highly accurate, non-invasive ways are preferable in the diagnosis of patients with possible elevated ICP in certain diseases such as idiopathic intracranial hypertension (IIH) or expansive lesions in the brain in conscious patients. With the continuous advancement of medical technology, non-invasive ICP measurement methods continue to develop and have several advantages, such as safety, reliability, low cost, convenience, and speed. Non-invasive ICP measurement has guiding significance for the early detection and treatment of neurological diseases and broad application prospects. Elevated ICP-induced blockage of CSF and blood circulation can lead to obvious pathological changes in the brain and fundus. Therefore, non-invasive measurement of ICP usually measures these physiological changes to evaluate ICP qualitatively or quantitatively.

Non-invasive ICP measurement methods comprise fluid dynamics, and ophthalmic, otic, electrophysiological, and other methods [8]. Among these methods, ocular measurement methods have attracted increased attention. Elevated ICP increases the pressure of the CSF in the subarachnoid space around the optic nerve sheath, thereby obstructing the circulation in the central retinal vein, which flows back through the retina and optic nerve head. The conditions inside the eye change as a result of this obstruction and result in signs such as papilledema, visual field defects, increased diameter of the optic nerve sheath, and increased intraocular pressure (IOP). Therefore, non-invasively measuring changes in ocular parameters to estimate ICP has a certain degree of reliability and accuracy. Compared with invasive ICP measurement, non-invasive ICP measurement is safe and can reduce the rate of complications. Non-invasive methods have low space/environmental and technical expertise requirements, good portability, and low cost. Non-invasive ICP measurement methods have great application potential for the triage of certain patients and for the diagnosis of diseases.

This article reviews the current methods of non-invasive measurement of ICP by ocular measurement, and evaluates the indications and main advantages and disadvantages. Figure 1 shows the locations of interest when measuring ICP by different ocular methods. At the end of the article, the future development of non-invasive ICP measurement technology is discussed.

## 2. Non-Invasive ICP Measurement

Non-invasive ICP measurement technology plays a vital role in large hospitals, community clinics, and other medical facilities. As ICP directly affects the optic nerve and eyeball owing to the direct communication between the intracranial and orbital subarachnoid spaces, the monitoring of ICP through ocular measurement is undoubtedly a reliable and promising research direction.

The ideal non-invasive method should have the following characteristics: convenient and simple operation, and less reliance on operators. Ideally, the method can continuously monitor and track the dynamic changes in ICP with high accuracy and can measure ICP quantitatively rather than qualitatively. Additionally, the ideal method would be less affected by a patient’s cardiovascular instability. Table 1 lists the methods mentioned in the article with the respective main characteristics. The general principles, advantages, disadvantages, indications, and applicable occasions of each technology will be introduced in detail below.

### 2.1. Optic Nerve Sheath Diameter (ONSD)

The optic nerve sheath (ONS) extends from the dura mater, and the subarachnoid space surrounding the optic nerve contains CSF that communicates with the intracranial CSF. When ICP is increased, intracranial CSF passes through the optic canal into the subarachnoid space, increasing the ONSD. With the maturity of medical imaging technology, the ONS can be imaged by ultrasonography, computed tomography (CT), and magnetic resonance imaging (MRI) (Figure 2). Among these methods, the indirect assessment of ICP by ultrasonographic measurement of ONSD has been proven safe, reliable, and non-invasive. This method can rapidly detect elevated ICP in only a few minutes.

Hayreh first suggested in 1968 that the CSF pressure in the intracranial subarachnoid space and the CSF pressure in the optic nerve sheath were consistent [10]. This provided the theoretical basis for later studies on the relationship between ONSD and ICP. Galetta et al. used ultrasonography to monitor ONSD and compared the values with those of ICP and found that the width of the ONSD increased as ICP increased [11]. Subsequently, several studies confirmed the existence of a linear relationship between ONSD and ICP [12,13,14]. One study found that changes in ONSD are synchronized to ICP changes [15]. The researchers corrected ICP in 19 patients with idiopathic IIH by therapeutic lumbar puncture. They found a significant and immediate reduction in ONSD and even a return to normal in half of the cases, indicating a rapid response of ONSD to changes in ICP [15]. Ocular ultrasonographic measurement of ONSD, as a portable and non-invasive method, can rapidly detect elevated ICP in patients with emergency traumatic brain injury. Most studies currently indicate an ONSD value of >5 mm as the threshold for determining elevated ICP, and researchers consider this diagnostic technique to have good accuracy [16,17,18].

Imaging of the ONS using CT or MRI can also evaluate patients for elevated ICP, but it takes longer to measure. In one study, the researchers performed simultaneous MRI and invasive ICP measurement in 38 patients with traumatic brain injury, and the results showed that ONSD measured by MRI was significantly positively correlated with ICP [19]. Some researchers have shown that MRI provides more accurate measurements compared with those of ultrasonography [20,21].

CT imaging of ONSD has similar sensitivity and specificity to those obtained with MRI in detecting elevated ICP [22]. Studies have found that ONSD also had a superior predictive value for increased ICP compared to classical CT findings of intracranial hypertension [23]. Additionally, the ONSD measurement on CT is highly reproducible and has a high potential for clinical application. However, conventional CT and MRI require patient transport to a dedicated radiology department, which is costly in both time and resources. Recent developments of medical imaging techniques such as portable CT [23] and MRI [24] allow more timely and accessible measurement of ONSD. For example, these portable devices could provide bedside measurement of ONSD for some severe TBI patients that have a low level of mobility.

Evaluation of ONSD is a relatively convenient and useful technique for detecting elevated ICP. ONSD may be used as one of the routine clinical judgment indicators. This method requires trained medical imaging technologist to operate, but the measurement results are less dependent on the operator. For ONSD ultrasonography, a study found that the sensitivity (95%) among trained operators seemed slightly superior to that of untrained operators (93%) [25]. ONSD is a good predictor of ICP for patients with IIH, TBI, hydrocephalus, and other diseases [18,26,27]. For TBI patients, ONSD, either quantified by ultrasound, MRI, or CT imaging, can easily differentiate the patient’s ICP. However, a study has found that ONSD is not suitable for patients with subarachnoid hemorrhage [28].

Additionally, differences in baseline ONSD [29] and ONS biomechanics [30] between individuals could lead to inaccurate estimation of ICP. Moreover, the inability to continuously monitor ICP using this method is also a disadvantage that cannot be ignored. Although this method has many advantages, the assessment of ICP using only the ONSD metric is not used much in clinical practice.

### 2.2. Flash Visual Evoked Potentials (FVEPs)

VEP accurately reflects visual pathway disorders and is one of the earliest and most well-studied cortical evoked potentials in clinical theory. According to the different forms of retinal stimulation, VEPs are divided into FVEP and graphic VEPs. Among these types, FVEP refers to the potential change in the occipital cortex after the retina is stimulated by a uniform flash of light, which can reflect the integrity of the visual pathway from the retina to the occipital cortex. The visual pathway is located at the base of the brain. Optic nerve dysfunction occurs often with intracranial lesions, and changes in the visual potential of the occipital lobe of the brain caused by photostimulation of the retina can reflect, to a certain extent, the pathophysiology of ICP. Mechanical compression of the brainstem and blood vessels induced by elevated ICP can impair cerebral blood circulation. This can cause ischemia and hypoxia of neurons and nerve fibers, resulting in subsequent impaired brain tissue metabolism and blocked neuronal electrical signal conduction. Under these conditions, the FVEP wave crest latency is prolonged, the wave amplitude is decreased, and the wave width is increased. This phenomenon is more obvious if brain herniation is present. Therefore, ICP can be measured indirectly by establishing the relationship between the latency of specific waves of FVEP and ICP.

As early as 1981, York et al. demonstrated a strong positive correlation between the increased N2 wave latency of VEPs and elevated ICP [31]. The researchers found a high predictive accuracy at ICP > 300 mm H_2_O. Subsequently, a study found that when patients with high ICP received mannitol to reduce ICP, the N2 wave latency was shortened, and there was a strong linear relationship with ICP (r = 0.97) [32]. Based on this phenomenon, Zhong et al. investigated the waveform extraction method for FVEP [33]. The researchers combined the advantages of the independent component analysis method, superimposed averaging method, and multi-resolution wavelet transform method to effectively acquire FVEP signals at the left and right occipital bones. Using the linear relationship between the N2 wave of FVEP and ICP, combined with the advantages of transcranial Doppler ultrasonography (TCD), these two methods were combined to develop a highly reliable and practical special instrument for non-invasive measurement of ICP, which overcomes the shortcomings of a single ICP measurement method.

Non-invasive ICP measurement devices developed based on the principle of FVEP have been used clinically [34] Generally, it only takes less than one minute to complete a single measurement. In clinical application, it needs to be operated by trained professionals. Normally, three measurements need to be completed within 15 min and the average value is used as the final measurement of ICP. Most clinical studies on FVEP non-invasive ICP monitoring technology focus on patients with craniocerebral trauma, subarachnoid hemorrhage, and hypertensive intracerebral hemorrhage-induced ICP elevation. This device may be used as one of the routine clinical examinations for patients with craniocerebral disease. However, a study found that FVEP did not change significantly in IIH patients [35]. Moreover, using FVEP to assess ICP is unsuitable in certain patients, such as those with frontal lobe hematoma, retinal damage, or optic neuropathy. In addition, factors such as blood glucose concentration, the patient’s nerve conduction rate, and electrolytes in the body can affect the measured ICP value using the FVEP assessment. In patients with severely elevated ICP, the accuracy of FVEP assessment is low.

### 2.3. Two-Depth Transorbital Doppler (TDTD) Ultrasonography

TCD is a non-invasive method for the measurement and evaluation of the hemodynamic parameters of the skull base arteries, and TCD is often used in the examination of various vascular diseases. When ICP continues to increase pathologically, cerebral blood flow decreases, cerebral perfusion pressure continues to decrease, and the dynamic parameters of arterial blood flow also change accordingly. Aaslid et al. observed that the mean blood flow velocity, systolic blood flow velocity, and diastolic flow velocity decreased when ICP increased, and the pulsatility index (PI) and resistive index increased significantly [36]. Predicting ICP based on PI is one of the most promising approaches in non-invasive ICP measurement as there is a strong linear relationship between PI and ICP [37,38]. However, the accuracy and reliability of predicting ICP based on PI remain controversial [39,40]. In one study, a multivariate linear regression model including hematocrit, mean arterial blood pressure, heart rate, and arterial carbon dioxide pressure showed that PI itself was not a good predictor of ICP [41]. Moreover, its prediction reliability was not significantly improved after adding other hemodynamic variables; thus, this method is unsuitable for clinical non-invasive ICP measurement.

Traditional TCD imaging is limited to scanning at a certain depth, and its reliability and reproducibility rely on manual operations. Additionally, there are differences in blood vessels between individuals; therefore, measuring the blood flow in a single arterial segment is insufficient to quantitatively determine ICP. To address these issues, TDTD was developed, which detects the blood flow spectrum of the extracranial and intracranial ophthalmic arteries by emitting high-frequency pulse waves of different frequencies. At the same time, external pressure is applied to the orbital tissue, to balance the influence of ICP on the intracranial ophthalmic artery. When the spectral waveform of the blood flow of the external and internal ophthalmic arteries becomes similar, it can be considered that the applied pressure is equal to the ICP [42]. The structure of the TDTD device is shown in Figure 3. The Vittamed 205 developed by Vittamed Corporation is a special device for measuring ICP using the TDTD principle. This equipment requires the operator to locate the extracranial and intracranial ophthalmic arteries with an ultrasound probe, and subsequent measurements can be automated with high accuracy. On average, it takes 16 min to complete the measurement [43]. This method applies to a wide range of scenarios, whether it is for the diagnosis of IIH patients, or rapid triage of TBI patients in emergencies (battlefield, ambulance, etc.). This method requires special training for operators, but may be widely used in the future.

TDTD is an important and reliable method for the non-invasive measurement of ICP. However, this method cannot be used for continuous monitoring. Recently, Lucinskas et al. proposed that continuous monitoring and analysis of ICP for up to 1 h could be achieved with this technique using the ophthalmic artery as the pressure sensor for ICP [45]. This technology has a certain application value for the diagnosis and treatment of some diseases that require continuous monitoring of ICP.

### 2.4. Central Retinal Venous Pressure (CRVP)

The arteries and veins in the retina are directly observable vessels in the human body. Therefore, shortly after the invention of the direct ophthalmoscope, Coccius used the device to detect spontaneous pulsation of the central retinal vein (CRV) [46]. Coccius observed that this pulsation was common and could be induced by increasing the IOP when spontaneous pulsation was not present. Later, other scholars discovered and confirmed that CRV pulsations can occur spontaneously in people with normal ICP and that CRV pulsations can be induced by increasing IOP through the compression of the eyeball [47]. However, in patients with elevated ICP, more pressure is required to trigger pulsation. This provides a non-invasive method for the early diagnosis of increased ICP [48,49].

In 1925, Baurmann used ophthalmodynamometry (ODM) to record the threshold force to induce venous pulsations and found a correlation between the threshold force and ICP; however, this finding received little attention [50]. Later, a study proposed that the minimum IOP required to induce retinal venous pulses was equal to the CRVP [51]. In response to this hypothesis, an experimental model that could simulate CRV collapse in vitro was established, thereby demonstrating that the venous collapse phenomenon enables the determination of CRVP [52]. After the CRV passes through the lamina cribrosa, the outside of the blood vessels in the optic nerve receives pressure from the CSF. To demonstrate the exact correlation between ICP and CRVP, Firsching et al. recorded ICP and CRVP in 22 patients and found a close linear relationship between these pressures (r = 0.983) [53]. The measurement process is shown in Figure 4. Another study by the same team with 102 subjects showed that the probability that CRVP correctly predicted elevated ICP was 84.2%. Additionally, the probability of normal CRVP predicting normal ICP was 92.8% [54]. A mathematical model also confirmed the feasibility of retinal vein pulses to predict CSF pressure [55].

Although it is feasible to predict ICP from CRVP, the accuracy of this method can be further improved by incorporating additional information. Querfurth et al. used a multivariate model combined with the results of central retinal artery Doppler ultrasonography blood flow measurement and CRVP to improve the accuracy of ICP prediction and invented a new ophthalmodynamometric method [57]. The application of this method requires the simultaneous use of ODM, an ophthalmoscope (or fundus camera), and a tonometer, which generally requires the joint operation of two operators. Therefore, it is rarely used in clinical practice. If these instruments can be integrated into a single device, it will promote the wide use of this method to rapidly differentiate high ICP subjects from normal subjects.

In 2020, Morgan et al. proposed a method of evaluating ICP using retinal vein photoplethysmography (PPG), in which the venous pulse amplitude is measured under different IOP levels to estimate the ICP/IOP balance point and hence, ICP [58]. The researchers then used this method to measure retinal vein pulsation amplitude and estimated ICP in 30 patients, finding that the technique appeared to be very useful in patients with IIH. This PPG technique appears well-suited for patients who are unsuitable for higher-risk measurement methods, such as lumbar puncture [59].

The CRVP method is suitable for situations that require low cost. Although the accuracy of this method needs further validation and improvement, CRVP and related physiological parameters are highly reliable in predicting ICP. Therefore, using CRVP to measure ICP has great clinical potential. The CRVP method of non-invasive measurement of the absolute value of ICP has been proven reliable and safe, but it also has limitations. For some patients, ICP is reduced rapidly by drainage but without resolution of the papilledema. CRVP remains higher than normal owing to the papilledema, which impedes retinal venous outflow. Therefore, the CRVP method is unsuitable for patients with persistent papilledema after a rapid decrease in ICP [53]. Additionally, the CRVP method cannot achieve continuous measurement of ICP, nor is it suitable for patients whose IOP is higher than the CRVP. However, this simple non-invasive measurement method still has great application value for the rapid triage of patients, especially for patients with traumatic brain injury.

### 2.5. Optical Coherence Tomography (OCT)

The optic nerve is mainly located intracranially, and changes in ICP may affect the anatomical structure of the intraocular optic nerve. OCT can provide reliable and reproducible quantitative measurements of changes in retinal structures in the area around the optic disc. Therefore, it has been suggested that OCT has diagnostic value for patients with intracranial hypertension.

Intracranial hypertension causes changes at the cellular or axonal level, creating swelling of the retinal nerve fiber layer (RNFL), the innermost layer of the retina. In 1998, Borchert et al. patented a method to estimate ICP using OCT measurements of RNFL thickness; however, the authors did not discuss the relationship between RNFL thickness and ICP [8]. Later, another study found that in patients with suspected IIH, in addition to the subjective assessment of papilledema, OCT can be used as an important supplement to the diagnosis [60]. However, OCT is of less value in patients with previously treated long-term IIH. OCT also has some limitations. For example, when papilledema is severe, OCT is of almost no value. In addition, one study found that papilledema could not be observed with an ophthalmoscope when ICP was elevated in some children with craniosynostosis [61]. However, OCT images showed a marked increase in RNFL thickness. It is believed that OCT enables a detailed examination of the optic nerve head and provides a potentially sensitive indicator of ICP elevation in craniosynostosis patients.

In 2011, Kupersmith et al. suggested that assessment of the peripapillary retinal pigment epithelium and Bruch’s membrane angle by OCT could be used for the measurement and qualitative assessment of papilledema [62]. The researchers used geometric morphometrics on OCT images in 41 patients with high ICP and found that the subsurface contour of the peripapillary retinal pigment epithelium–basement membrane (ppRPE/BM) layer changed with decreasing ICP (Figure 5). This finding can be used as an adjunct to the assessment of ICP with RNFL thickness [63]. Although an increase in CSF deforms the ppRPE/BM layer and the subjacent sclera toward the vitreous, it is difficult to evaluate the change in ICP only based on this condition. Recent studies showed that the spatial configuration of ppRPE/BM varies with age and ocular diseases. With increasing age, the angle between the nasal and temporal ppRPE/BM changed from an inverted V-shape to a more pronounced V-shaped configuration [64]. Similarly, V-shaped configurations are more pronounced in glaucoma patients compared with healthy subjects [65]. Therefore, this method can be used as a supplementary measurement for ICP evaluation.

Recently, a study proposed that OCT could be used for both static ICP estimation as well as for the assessment of pulsatile ICP [66]. The researchers used a line connecting both sides (nasal-temporal) of the scleral canal opening as a reference line, and the measured height from the highest point of the optic nerve head to the reference line was called the optic nerve head height. The study found that the peripapillary Bruch’s membrane angle and optic nerve head height correlated with the pulsatile ICP, thereby allowing assessment of whether ICP is elevated. This study provided a non-invasive method for detecting abnormal pulsatile ICP [66].

As a new type of tomography technology, OCT has an irreplaceable role in the diagnosis of ophthalmic diseases, and it is also a potential tool for ICP assessment in IIH patients. However, OCT can assess the level of ICP only qualitatively and cannot provide an accurate value. Additionally, the method is unsuitable for patients with severe papilledema. The diagnosis of ICP by OCT is unsuitable for clinical dissemination; however, the method is viable for ophthalmologists to diagnose patients with IIH.

### 2.6. Pupillometry

Some studies have preliminarily proved that changes in pupil diameter and light reflection can reflect changes in ICP [67,68]. If an impaired pupillary light reflex is detected, this may indicate an increase in ICP. This finding is owing to increased ICP causing mechanical compression of the pathway of the oculomotor nerve, which inhibits pupillary reactivity. Traditional pupil size measurement uses a ruler for visual measurement; however, with this method, it is impossible to measure small pupillary contractions or continuous changes. As technology has continued to develop, automatic pupillometers based on infrared technology have emerged. Automatic pupillometers are more accurate and reliable than manual examinations for measuring pupil size and reactivity, and can also measure subtle changes in the pupillary light response.

Pupil change is an important sign of condition change in patients with craniocerebral injury. One study found that pupil diameter in healthy people decreased by an average of 34% under light stimulation, while the pupil diameter of head trauma patients decreased by an average of 20% [69]. These results suggest that changes in pupil diameter measured by a pupillometer can reflect changes in ICP.

The Neurological Pupil index (NPi) is derived by algorithmically transforming the seven parameters involved in the pupillary light reflex. The NPi is used to quantify pupillary reactivity. The original developers of the NPi used this index to assess pupillary reactivity and found that patients with an abnormal pupillary light response had ICP values far above the normal value [70]. This suggests that quantitative measurement and classification of pupillary reactivity using the NPi may be useful in the early determination of increased ICP. However, one study found that elevated ICP resulted in lower NPi values and contraction velocity, with no effect on pupil size [71]. Additionally, there was a weak but not statistically significant relationship between NPI and ICP. Therefore, pupillometry may not be a reliable predictor of ICP [72]. Although automated pupillometry does not appear to be adequate for assessing ICP, this method may be useful for identifying patients with intracerebral hemorrhage without elevated ICP [73].

Whether pupillometry can be used for non-invasive measurement of ICP remains controversial. However, it is certain that pupillary reactivity is affected by many factors, namely several neurological disorders, various medications, the emotional state of the individual, and the time of day. These factors limit the application of pupillary reactivity evaluation; however, the method is still useful in determining ICP in patients with severe craniocerebral injury.

### 2.7. IOP Measurement

Measurement of IOP also appears to be a viable method for assessing ICP in patients with known intracranial lesions. Increased venous pressure in the cavernous sinus is transmitted to the episcleral vein through the superior ophthalmic vein, and elevated episcleral venous pressure leads to increased IOP. Therefore, IOP increases as ICP increases. Some studies have found that patients with abnormal ICP also have abnormal IOP, and that there is a strong correlation between the two pressures [74,75]. Therefore, IOP may be a good indicator of abnormal ICP in patients with known intracranial lesions. However, subsequent studies found completely opposite results, indicating that there is no correlation between IOP and ICP, and that IOP measurement cannot effectively replace ICP measurement [76,77].

Although a correlation between IOP and ICP has been found in many animal and human studies, IOP is influenced by many factors. Patients with glaucoma, ocular hypertension, and other diseases can have abnormal IOP. Therefore, clinically, it is difficult to determine whether high IOP is caused by intracranial hypertension or ocular diseases. Additionally, there is no specific threshold value to determine whether ICP is abnormally elevated owing to variations in baseline IOP between individuals and the disagreements between various IOP measurement techniques. Therefore, IOP is not a reliable parameter for ICP assessment.

Pupillary reactivity evaluation and IOP measurement are basic examinations in ophthalmology clinics, which are easy to operate and fast to examine. Although it cannot be used as an indicator for detecting abnormal ICP alone, it provides a possible solution for the screening of certain craniocerebral diseases.

### 2.8. Retinal Arteriole and Venule Diameter Ratio (A/V Ratio)

Elevated ICP can lead to dilated retinal venules [51,78]. Therefore, the ratio of the diameter of the retinal arterioles and venules is correlated with ICP. As fundus imaging is relatively easy and inexpensive, it is possible to automate this method. In a recent study by Andersen et al. [79], the ICP of 14 subjects increased after the injection of isotonic sterile saline by lumbar puncture. Continuous fundus imaging and ICP measurement were conducted during the injection, and the A/V ratio was automatically calculated through a deep learning algorithm. The result showed that when the average ICP value was higher than 15 mmHg, there was a negative linear relationship between the ICP increase and the A/V ratio. However, when the ICP value was lower than 15 mmHg, there was no correlation between the ICP and the A/V ratio. This method is expected to allow for fully automated non-invasive assessment of high ICP and may be able to measure the absolute value of ICP. However, whether this method can be applied clinically requires further research. If a fully automatic measurement device based on A/V ratio can be developed, it may be possible to realize self-measurement of patients at home. This is of great value for some patients who require daily monitoring of ICP.

## 3. Conclusions

Currently, several non-invasive ICP measurement methods have been proposed and verified. However, at present, none of the non-invasive methods have shown sufficient accuracy and reliability for use in daily clinical practice. A few methods have been shown to be promising although further validations and developments are needed. These non-invasive methods are not developed to replace invasive methods but could be useful for rapid triage, specific patients that are not suitable for invasive methods, or occasions when invasive methods are not available.

## Figures and Tables

**Figure 1 bioengineering-09-00304-f001:**
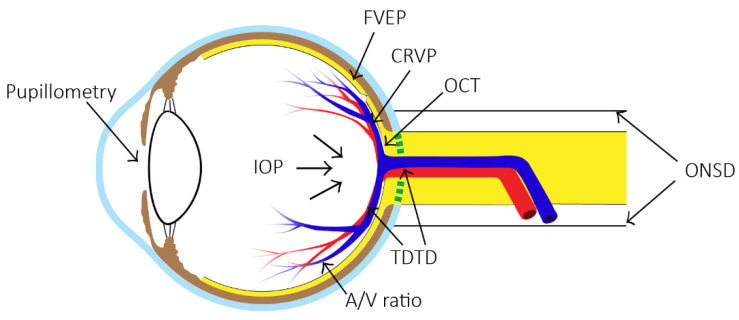
Eight methods of non-invasive measurement of ICP at their respective positions on the eye. (ONSD = optic nerve sheath diameter, FVEP = flash visual evoked potentials, TDTD = two-depth transorbital Doppler ultrasonography, CRVP = central retinal venous pressure, OCT = optical coherence tomography, IOP = intraocular pressure measurement, A/V ratio = retinal arteriole and venule diameter ratio).

**Figure 2 bioengineering-09-00304-f002:**
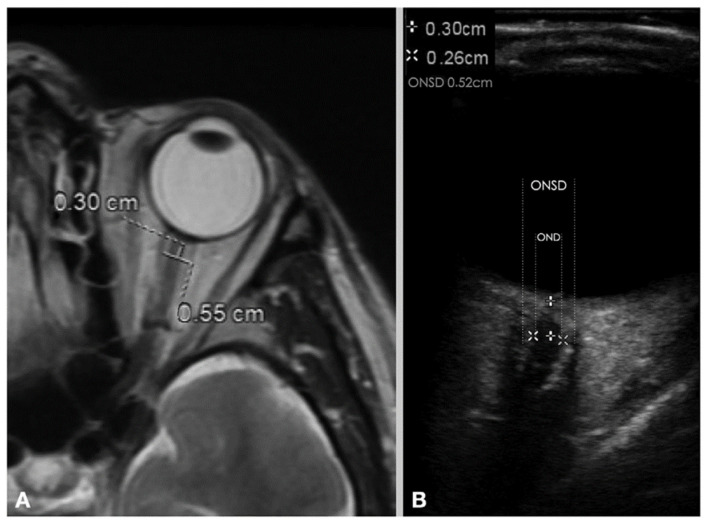
Measurement of ONSD and OND with (**A**) MRI image and (**B**) ultrasonography image. (ONSD = optic nerve sheath diameter, OND = optic nerve diameter). (Reprinted with permission from Ref. [9]. Copyright © 2020, Wiley).

**Figure 3 bioengineering-09-00304-f003:**
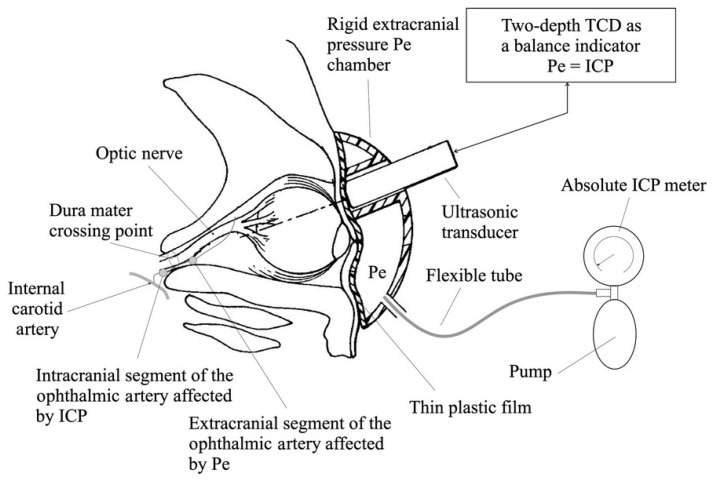
TDTD device for absolute ICP measurements. The ultrasound transducer of the Doppler subsystem is surrounded by an externally applied pressure chamber with a controlled external pressure source to balance the influence of ICP on the intracranial ophthalmic artery. (Reprinted with permission from Ref. [44]. Copyright © 2015, Siaudvytyte, L. et al.).

**Figure 4 bioengineering-09-00304-f004:**
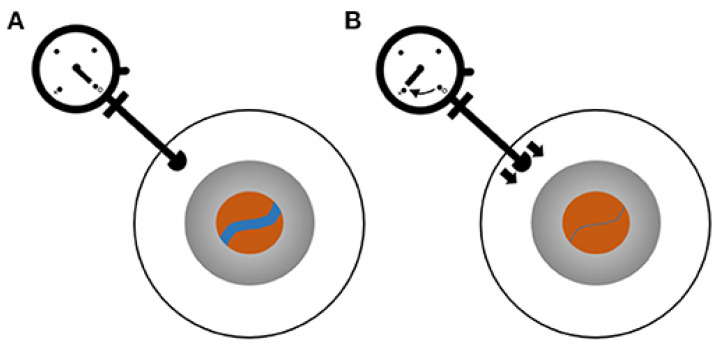
ODM for the measurement of CRVP. Blue line indicates CRV. (**A**) Baseline status with no force applied to the eye. (**B**) Pressure is applied by pushing the ophthalmometer against the sclera until the CRV collapses. (Reprinted with permission from Ref. [56]. Copyright © 2021, Moss, H.E.).

**Figure 5 bioengineering-09-00304-f005:**
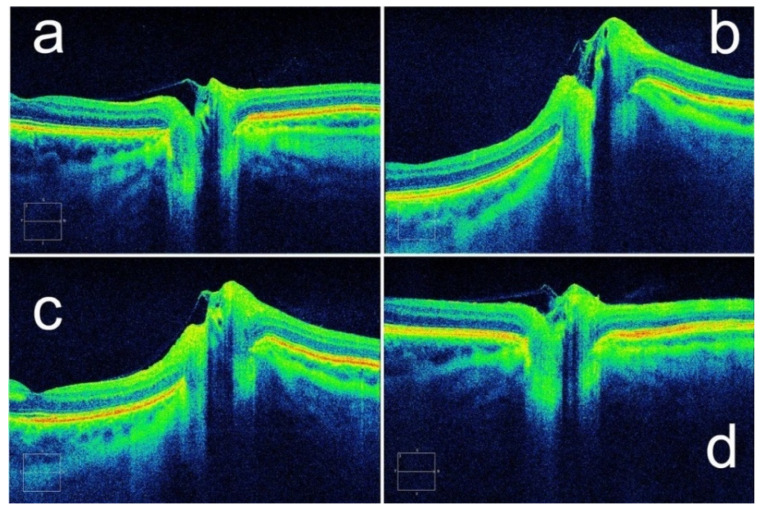
A patient’s ppRPE/BM layer shape changes with ICP. Configurations of ppRPE/BM layer at (**a**) baseline ICP, (**b**) ICP of 35 cm H_2_O, and (**c**) ICP of 27 cm H_2_O. (**d**) After a ventriculoperitoneal shunt was placed, the ppRPE/BM layer shape returned to baseline. (Reprinted with permission from Ref. [63]. Copyright © 2014, Assoc Research Vision Ophthalmology Inc.).

**Table 1 bioengineering-09-00304-t001:** Comparison of Methods for Non-invasive Measurement of ICP by Ocular Measurement.

Method	Equipment	Continuous Monitoring	Quantitatively	Comments
Optic Nerve Sheath Diameter (ONSD)	Ultrasonography/Computed Tomography/Magnetic Resonance Imaging	No	No	Not accurate due to differences in baseline ONSD between individuals
Flash Visual Evoked Potentials (FVEP)	Electroencephalo-gram	No	No	Not suitable for certain patients, such as those with frontal lobe hematoma, retinal damage, or optic neuropathy
Two-depth Transorbital Doppler (TDTD)	Transcranial Doppler Ultrasonography	No/Yes	Yes	Operators need professional training
Central Retinal Venous Pressure (CRVP)	Ophthalmodyn-amometer	No	Yes	Not suitable for patients with persistent papilledema after a rapid decrease in ICP
Optical Coherence Tomography (OCT)	OCT	No	No	Not suitable for patients with severe papilledema
Pupillometry	Automatic pupillometer	No	No	Pupillary reactivity is affected by many factors
IOP	Tonometer	No	No	IOP is not a reliable parameter for ICP assessment
Retinal Arteriole and Venule Diameter Ratio	Fundus camera	No	No	Accuracy still requires further research

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
