# Peer review of "A Review of the Methods of Non-Invasive Assessment of Intracranial Pressure through Ocular Measurement"

_bioengineering, 2022, doi:10.3390/bioengineering9070304_

Round 1
Reviewer 1 Report
The authors have written a review of non-invasive intracranial pressure monitoring. The general principles of different technologies are reviewed, but the specifics of each (advantages, disadvantages, indications, practical use) are loosely stated. A more comprehensive review should include answers to the following questions:
1. How often are these technologies used clinically?
2. How long does it take to measure ICP with each technique?
3. What special equipment is used for each?
4. What special training is needed to be able to use the technologies?
5. Who can use the technology to measure ICP?
6. How practical is use of the technology in a clinical (routine and emergent) setting?
7. What are the specific indications for each technology, and what is the best technology to use for those technologies with multiple indications? For instance, which technologies could be used to determine ICP in a child with craniosynostosis, and which technology would be best to use?
Several other concerns follow:
1. In line 139-140, the authors should note that portable CT is currently available. Are the specifications of these devices sufficient to determine optic nerve sheath diameter?
2. The CONCLUSIONS are too long and contain new information. The authors should shorten the CONCLUSIONS to 3 or 4 sentences which summarize their review. New material included in the current CONCLUSIONS (such as table 1) should be move to an appropriate existing section or a new section.
Author Response
Point 1: The authors have written a review of non-invasive intracranial pressure monitoring. The general principles of different technologies are reviewed, but the specifics of each (advantages, disadvantages, indications, practical use) are loosely stated. A more comprehensive review should include answers to the following questions:
Response 1: We very much appreciate the thorough review and thoughtful comments and suggestions made by the reviewer. We have worked diligently to respond to each suggestion as outlined below.
Point 2: How often are these technologies used clinically?
Response 2: Invasive intracranial pressure (ICP) measurement is still the most commonly used method. In clinical practice, the usage of non-invasive methods are not common. There are many clinical studies using ultrasound, MRI, or CT to quantify optic nerve sheath diameter (ONSD) for the estimation of ICP, which has been shown to be able to differentiate high and low ICP. Although frequently used in research, the assessment of ICP using only the ONSD metric is not used much in clinical practice. At present, commercial devices for non-invasive measurement of ICP using flash visual evoked potentials (FVEP) and two-depth transorbital Doppler (TDTD) technologies have been developed and used in clinical practice. However, it should be noted that these devices are not widely available in hospitals. As a relatively new technology, TDTD has great clinical potential, and many studies have shown that this method is worthy of further development for widely clinical application. Using central retinal venous pressure (CRVP) to assess ICP requires multiple ophthalmic examination devices and two operators’ cooperation. Therefore, it is rarely used in clinical practice. However, if these ophthalmic devices can be integrated into a single device, it will promote the wide use of this method for rapid pre-diagnosis triage of patients. Optical coherence tomography (OCT), pupillometry, and intraocular pressure (IOP) are rarely used directly in clinical practice and can only be used as indicators for auxiliary diagnosis. As a new method, retinal arteriole and venule diameter ratio (A/V ratio) has the potential to be automatically measured as it only relies on fundus images. However, at present, A/V ratio remains in the research stage though it has particular clinical potential in the future.
We have added these contents to each section accordingly. (section 2.1, 2.2, 2.3, 2.4, 2.5, 2.7, 2.8)
Point 3: How long does it take to measure ICP with each technique?
Response 3: Using ultrasonography to monitor ONSD for detection of elevated ICP only requires a few minutes. However, measurement of the ONSD using CT or MRI takes a longer time depending on the slice thickness and scanning protocol but normally, they can be finished within half an hour. For patients with traumatic brain injury (TBI), MRI or CT examination is necessary. This allows clinicians to use ONSD to assess the level of ICP with these head scans. As for TDTD, a study shows that the average measurement time of operators is 16 minutes using dedicated devices. In addition, other methods mainly rely on the measurement speed of ophthalmic devices, which generally only take a few minutes.
We have added these contents in our revised manuscript. (section 2.1, 2.2, 2.3, 2.7)
Point 4: What special equipment is used for each?
Response 4: As discussed in the original manuscript, ONSD can be measured by ultrasonography, CT, and MRI. At present, commercial devices using the principle of FVEP has been developed and used in clinical practice. Similar to FVEP, TDTD-based commercial devices such as Vittamed 205 developed by Vittamed Corporation are available. Other methods can be achived using only basic ophthalmic equipments.
We have added these information in the revised manuscript. (section 2.2, 2.3)
Point 5: What special training is needed to be able to use the technologies?
Response 5: Imaging of the optic nerve with ultrasound, CT and MRI uses standard imaging equipments, thus corresponding trainings are needed to be able and eligible to operate these devices. Measurement of ONSD from these medical images are straightforward with low operator dependency. One study found that, using ultrasonography, sensitivity (95%) among trained operators seemed slightly superior to that of untrained operators (93%). TDTD equipment requires the operator to locate the extracranial and intracranial ophthalmic arteries with an ultrasound probe, and subsequent measurements can be automated. The remaining methods are mainly based on basic ophthalmic equipments. Therefore, relevant trainings for ophthalmic technicians and ophthalmologists are needed.
We have added these information in our revised manuscript. (section 2.1, 2.2, 2.3)
Point 6: Who can use the technology to measure ICP?
Response 6: These methods currently require healthcare professionals with certifications to operate. As A/V ratio relays only on fundus images, it is possible that A/V ratio can be democratized and used by people at home with dedicated devices or even smartphones.
We have added these contents to the revised manuscript. (section 2.1, 2.8)
Point 7: How practical is use of the technology in a clinical (routine and emergent) setting?
Response 7: Bedside ultrasound measurement of ONSD is a well-studied method that can quickly determine the level of ICP. MRI or CT are routine examinations for patients with traumatic brain injury and ONSD can be obtained from these scans. Studies have found that ONSD had a superior predictive value for increased ICP compared to classical CT brain image findings of intracranial hypertension. ONSD may be used as one of the routine clinical examinations. FVEP and TDTD technologies can be used for regular clinical examinations, and they are more convenient to apply as commercial devices are available. TDTD equipment is small and portable thus can be used on occasions requiring rapid triage or preliminary screening, such as ambulances, emergency room, community outpatient clinics, etc. OCT, pupillary and IOP measurement can be performed routinely though these three methods are far from accurate for measuring ICP. However, they provide a basis for doctors to find potential patients with abnormal ICP. Evaluating ICP by measuring CRVP is more complicated. However, as mentioned above, if a dedicated device can be developed, it is very suitable for emergency triage situations, such as battlefields and emergency room. The same is true for the fundus-based A/V ratio. On the premise of proving the accuracy of this method, its advantages of fully automatic evaluation are suitable for large-scale screening.
We have added the relevant content separately to the corresponding positions in the text. (section 2.1, 2.2, 2.3, 2.4, 2.5, 2.7, 2.8)
Point 8: What are the specific indications for each technology, and what is the best technology to use for those technologies with multiple indications? For instance, which technologies could be used to determine ICP in a child with craniosynostosis, and which technology would be best to use?
Response 8: For patients with severe TBI, invasive methods are often used to monitor ICP. At the same time, many studies have shown that ONSD, FVEP, TDTD, and other techniques are helpful for the rapid triage of TBI patients. ONSD, either measured by ultrasound, MRI, or CT imaging, can easily differentiate patients with ICP elevation.
Idiopathic intracranial hypertension (IIH) can induce dizziness and nausea in patients. CT or MRI imaging and optic nerve head examinations are essential for the diagnosis of IIH patients. Therefore, non-invasive ICP measurement based on these examinations are possible to make a rough distinction between patients with increased ICP and normal subjects. However, the change in FVEP in these patients is not apparent. Specialized TDTD equipment has little harm to patients and high portability. Therefore, it is a promising method for evaluating ICP in patients with IIH.
Ultrasound measurement of ONSD is not an appropriate method for assessing ICP in children with craniosynostosis. A study found that ONSD ultrasonography showed a sensitivity of only 11% in patients with craniosynostosis. However, OCT of the retina produces a potentially sensitive indicator of ICP in craniosynostosis patients.
Among these methods, ONSD and TDTD have a wide range of applications and can be used for preliminary ICP estimation for most diseases with an elevated ICP. However, some studies have found that ONSD is not suitable for patients with subarachnoid hemorrhage.
We have added these discussions in the revised manuscript. (section 2.1, 2.2, 2.3, 2.4, 2.5)
Point 9: In line 139-140, the authors should note that portable CT is currently available. Are the specifications of these devices sufficient to determine optic nerve sheath diameter?
Response 9: Thank you, this is a very good point. The specifications of portable CT is enough to image the optic nerve, which has already been used to measure ONSD for ICP estimation. Moreover, mobile portable CT (or MRI) allows more timely and accessible measurement of ONSD. For example, these portable devices could provide bedside measurement of ONSD for severe TBI patients that have a low level of mobility.
We have modified the text in the ONSD section accordingly (line 150-158):
“……However, conventional CT and MRI require patient transport to a dedicated radiology department, which is costly in both time and resources. Recent developments of medical imaging techniques such as portable CTand MRI allow more timely and accessible measurement of ONSD. For example, these portable devices could provide bedside measurement of ONSD for some severe TBI patients that have a low level of mobility. ”
Point 10: The CONCLUSIONS are too long and contain new information. The authors should shorten the CONCLUSIONS to 3 or 4 sentences which summarize their review. New material included in the current CONCLUSIONS (such as table 1) should be move to an appropriate existing section or a new section.
Response 10: We thank the reviewer for this constructive comment. As suggested, we have shortened the conclusions section and moved part of the texts in the original conclusion and the table to the introduction section (line 97-111). Moreover, we also moved texts on the CRVP method to section 2.4.
Reviewer 2 Report
This review of noninvasive methods for checking ICP through the orbit is clear, simple and concise, giving a broad spectrum of the possibilities and their limitations. What worries me is that although there is still no possibility to replace invasive methods for monitoring ICP, the authors stress the complications and "limited range of applications" of invasive ICP monitoring, which constitutes the gold standard up till now. I don't see the utility of limiting invasive methods while you can not offer a real alternative.
Most of the noninvasive methods proposed are of limited value by many different reasons, mostly because they are not applicable to continuous monitoring nor quantification of the findings. Their applicability is mostly based on discriminating patients with possible elevated ICP in the context of some pathologies as IIH or expansive lesions in the brain in conscious patients.
Therefore, I believe the paragraph number 2 (invasie ICP monitoring) should be suppressed at the same time that the "Introduction" paragraph should stress the concept of invasive CSF monitoring as the gold standard of ICP monitoring.
It is not against this invasive monitoring of critically iull patients that non-invasive methods are being investigated, but because there are many clinical conditions that may induce elevations of ICP and could be potentially diagnosed and even controlled by non invasive methods.
I would recommend to present this review as an updated revision of the noninvasive methods of measuring ICP through the orbit, presenting the current indications / limitations, and the need of future research.
In no case I believe these techniques will by now replace the invasive ICP monitoring in neurocritical care and this paper should avoid comparing these techniques with different and still limited applications. This conclusion should be clearly stressed to avoid confusion.
Author Response
Point 1: This review of noninvasive methods for checking ICP through the orbit is clear, simple and concise, giving a broad spectrum of the possibilities and their limitations. What worries me is that although there is still no possibility to replace invasive methods for monitoring ICP, the authors stress the complications and "limited range of applications" of invasive ICP monitoring, which constitutes the gold standard up till now. I don't see the utility of limiting invasive methods while you can not offer a real alternative.
Most of the noninvasive methods proposed are of limited value by many different reasons, mostly because they are not applicable to continuous monitoring nor quantification of the findings. Their applicability is mostly based on discriminating patients with possible elevated ICP in the context of some pathologies as IIH or expansive lesions in the brain in conscious patients.
Response 1: We are very grateful to the Reviewer for these constructive comments. We agree that invasive ICP monitoring as the gold standard is crucial for many critically ill patients, especially those with severe traumatic brain injury. Invasive measurements are not only highly accurate but also able to monitor ICP continually. These non-invasive methods are not developed to replace invasive methods but could be useful for rapid triage, specific patients that are not suitable for invasive methods or occasions when invasive methods are not available.
As suggested, we have included the following text to the Introduction section, emphasizing the irreplaceability of invasive intracranial pressure monitoring (line 52-59).
“Although invasive ICP monitoring requires dedicated space/environment and is prone to associate with some complications, it still has an irreplaceable role for many patients. For example, for patients with severe craniocerebral injury, dynamic ICP monitoring is an important indicator to determine or adjust treatment plans. Because it is difficult to achieve ideal accuracy and technological requirements for continuous monitoring with non-invasive ICP measurement equipment, invasive ICP measurement is still the first choice for monitoring ICP in patients with severe traumatic brain injury (TBI).”
We have also added the following sentence to the conclusion section:
“These non-invasive methods are not developed to replace invasive methods but could be useful for rapid triage, specific patients that are not suitable for invasive methods or occasions when invasive methods are not available.”
Point 2: Therefore, I believe the paragraph number 2 (invasie ICP monitoring) should be suppressed at the same time that the "Introduction" paragraph should stress the concept of invasive CSF monitoring as the gold standard of ICP monitoring.
Response 2: Thank you. We have deleted the section on invasive ICP measurement and moved these contents to the introduction section. As mentioned above, we have stressed the concept of invasive CSF monitoring as the gold standard of ICP monitoring in the introduction section.
We have made these changes in the revised manuscript (line 39-59).
Point 3: It is not against this invasive monitoring of critically iull patients that non-invasive methods are being investigated, but because there are many clinical conditions that may induce elevations of ICP and could be potentially diagnosed and even controlled by non invasive methods.
Response 3: We agree with you that non-invasive methods are not developed to replace invasive methods. However, in some special occasions, such as ambulances, emergency room or battlefields, where rapid triage is required, non-invasive measurements are of great value for diagnose of elevated ICP. We have made a more detailed description of the suitable occasions and indications of each method in the revised manuscript (section 2.1, 2.2, 2.3, 2.4, 2.5, 2.7, 2.8).
Point 4: I would recommend to present this review as an updated revision of the noninvasive methods of measuring ICP through the orbit, presenting the current indications / limitations, and the need of future research.
Response 4: We thank the reviewer for this constructive comment and we have modified the manuscript accordingly as suggested. Current indications and limitations of each methods are discussed.
Point 5: In no case I believe these techniques will by now replace the invasive ICP monitoring in neurocritical care and this paper should avoid comparing these techniques with different and still limited applications. This conclusion should be clearly stressed to avoid confusion.
Response 5: We agree with the reviewer that non-invasive methods cannot and should not be replaced at present. As suggested, we have further clarified this point in the introduction and conclusion sections (see the response for point 1).
Round 2
Reviewer 1 Report
The authors have satisfactorily addressed the reviewer's comments.
Reviewer 2 Report
I thank the author's effort to adjust the suggested changes and comments.
I believe the revised manuscript has won in clarity and applicability while adjusting to real expectations on these methods of noninvasively measuring ICP.